# Imbalanced Loss-Integrated Deep-Learning-Based Ultrasound Image Analysis for Diagnosis of Rotator-Cuff Tear

**DOI:** 10.3390/s21062214

**Published:** 2021-03-22

**Authors:** Kyungsu Lee, Jun Young Kim, Moon Hwan Lee, Chang-Hyuk Choi, Jae Youn Hwang

**Affiliations:** 1Information and Communication Engineering, Daegu Gyeongbuk Institute of Science and Technology, Daegu 42988, Korea; ks_lee@dgist.ac.kr (K.L.); moon2019@dgist.ac.kr (M.H.L.); 2The Department of Orthopedic Surgery, School of Medicine, Catholic University, Daegu 42472, Korea; drjunyoung@cu.ac.kr

**Keywords:** rotator-cuff tear, semantic segmentation, deep learning

## Abstract

A rotator cuff tear (RCT) is an injury in adults that causes difficulty in moving, weakness, and pain. Only limited diagnostic tools such as magnetic resonance imaging (MRI) and ultrasound Imaging (UI) systems can be utilized for an RCT diagnosis. Although UI offers comparable performance at a lower cost to other diagnostic instruments such as MRI, speckle noise can occur the degradation of the image resolution. Conventional vision-based algorithms exhibit inferior performance for the segmentation of diseased regions in UI. In order to achieve a better segmentation for diseased regions in UI, deep-learning-based diagnostic algorithms have been developed. However, it has not yet reached an acceptable level of performance for application in orthopedic surgeries. In this study, we developed a novel end-to-end fully convolutional neural network, denoted as Segmentation Model Adopting a pRe-trained Classification Architecture (SMART-CA), with a novel integrated on positive loss function (IPLF) to accurately diagnose the locations of RCT during an orthopedic examination using UI. Using the pre-trained network, SMART-CA can extract remarkably distinct features that cannot be extracted with a normal encoder. Therefore, it can improve the accuracy of segmentation. In addition, unlike other conventional loss functions, which are not suited for the optimization of deep learning models with an imbalanced dataset such as the RCT dataset, IPLF can efficiently optimize the SMART-CA. Experimental results have shown that SMART-CA offers an improved precision, recall, and dice coefficient of 0.604% (+38.4%), 0.942% (+14.0%) and 0.736% (+38.6%) respectively. The RCT segmentation from a normal ultrasound image offers the improved precision, recall, and dice coefficient of 0.337% (+22.5%), 0.860% (+15.8%) and 0.484% (+28.5%), respectively, in the RCT segmentation from an ultrasound image with severe speckle noise. The experimental results demonstrated the IPLF outperforms other conventional loss functions, and the proposed SMART-CA optimized with the IPLF showed better performance than other state-of-the-art networks for the RCT segmentation with high robustness to speckle noise.

## 1. Introduction

A rotator cuff tear (RCT) is a common occurrence that can lead to pain, weakness, and limited range of motion in the shoulder joint [1,2,3]. In severe cases of RCT that can cause a functional disability, an accurate diagnosis and treatment are necessary. Symptomatic RCTs can be treated either nonsurgically or surgically. The precise diagnosis of RCT is essential before surgical treatment [4,5,6].

Among non-invasive imaging techniques, ultrasound imaging (UI) and magnetic resonance imaging (MRI) systems have been widely used for the diagnosing locations of RCT [3,6]. An MRI system has been a favorable imaging tool for localizing an RCT but it has a few limitations [7,8]; e.g., it is not easily accessible owing to its high cost, acoustic noise, and time consumption, and it cannot be applied to the diagnosis of locations of RCT dynamically. However, USG also has several advantages compared to MRI, including, a real-time dynamic capture, wide availability, low-cost, and time efficiency. Therefore, UI can be an alternative to MRI for the diagnosis of locations of RCT [8,9,10].

Various image processing-based segmentation techniques for the diagnosis of locations of RCT have been developed with the advancement of computing power [11]. Conventional image processing algorithms based on morphological operations such as dilation and erosion have been applied to the diagnosis of locations of RCT, although their performance needs to be further improved [7,8,12]. Contour-based segmentation methodologies have recently been used as an alternative solution for the segmentation of regions of an RCT [3,13]. However, they have also shown a limited performance in the diagnosis of locations of RCT owing to the long processing time required to produce the results, thus hindering the rapid translation of the techniques in clinics for an RCT diagnosis.

More recently, several deep-learning algorithms for physical image analysis have been introduced. In particular, deep-learning-based segmentation algorithms have shown a superior performance in the segmentation of diseased regions compared to conventional image processing algorithms [14,15]. A deep-learning algorithm with a channel attention module with multi-scale average pooling has been developed for better segmentation of breast tumors [14]. Li et al. [15] applied a deep-learning-based image processing algorithm as a diagnostic tool for the heart. The deep-learning-based diagnostic analysis outperformed the conventional image processing techniques for the diagnosis of various diseases.

However, conventional deep-learning-based segmentation algorithms may not be suited for the segmentation of RCTs in ultrasound images because of the following reasons: (1) an ultrasound image typically has a low contrast and high speckle noise [16]; (2) RCT regions in an ultrasound image are much smaller (typically less than 10%) than non-diseased areas. Please note that the skewed distribution of RCT in an ultrasound image may lead to poor predictive performance in the deep-learning-based analysis because such an imbalanced dataset forces the pixel-wise classification algorithm to be biased to the majority class, i.e., the sparse number of pixels, which indicate a specific class, can be considered to be imbalanced data distributions for the segmentation tasks; (3) In addition, the low contrast and noisy environment in the segmentation of ultrasound images can degrade the segmentation performance of deep-learning-based algorithms. Therefore, an advanced deep-learning algorithm further needs to be developed for the better diagnosis of RCT in ultrasound images [17,18,19].

Therefore, in this study, we propose a convolutional neural network (CNN)-based deep-learning architecture, denoted as a Segmentation Model Adapting pRe-Trained Classification Architecture (SMART-CA), for an accurate segmentation of RCT in USG. Here, in addition to a trainable encoder (normal encoder), we implemented a pre-trained encoder [20] into SMART-CA to detect the location of the RCT prior to the RCT segmentation. Furthermore, we developed a novel integrated-on-positive-loss function (IPLF) to mitigate the issues caused by an imbalanced dataset such as the RCT dataset in the optimization of the SMART-CA. In this study, we examine whether the IPLF enhances the performance of the SMART-CA in the segmentation of an RCT in an imbalanced dataset. In addition, we compare the SMART-CA with other deep-learning state-of-the-art and conventional handcrafted models to assess the performance of the SMART-CA for the segmentation of RCT, thus demonstrating the potential of the SMART-CA for a better diagnosis of locations of RCT using USG and an analysis in a clinical setting.

## 2. Methods

### 2.1. Dataset

An ultrasound imaging (UI) system for the diagnosis of locations of RCT was constructed in a previous study [3]. Clinical experiments were conducted on 35 patients who had RCTs of different sizes (massive, large, medium, and small) and states using the constructed system. A total of 1400 ultrasound images of both shoulders were acquired to generate datasets for both the RCT and healthy shoulder. Among 1400 ultrasound images, the number of ultrasound images of the RCT shoulder is 1280 while the number of ultrasound images of the healthy shoulder was 120. The ultrasound images from patients were randomly shuffled to construct datasets for the training and testing of the deep learning models. The images from 21 patients were utilized for a training set, the images from 7 patients were used as a validation set, and the images from another 7 patients were used as a test set. The training, validation and test sets included 840, 280, and 280 images respectively.

Figure 1 shows the samples of acquired images. To annotate the datasets, the acquired US images were devided into two classes: Class-0 and Class-1. Class-1 indicates the regions of RCTs which include all objects except for the RCT regions. Annotations were confirmed by two professional orthopedic doctors. However, the datasets for RCTs are imbalanced because the regions of the RCTs were small within the field of view (FoV) of the ultrasound images. Figure 1 shows the distribution of the RCT regions corresponding to the US images. This demonstrates that the dataset is imbalanced with the small area of the RCTs. The term sparse is utilized to indicate the small number of pixels in RCTs whose area is smaller than the background area, and thus the RCT (Class-1) is a sparse set of hard examples [21]. Therefore, the imbalanced training with an imbalanced dataset can degrade the performance of the deep learning algorithm in the segmentation of the RCT since the optimization of the network is biased toward one class [21,22]. Since the number of pixels of the background are much higher than that of RCTs, Class-0 is considered to be an easy case while Class-1 is considered to be a difficult case [21].

### 2.2. Integrated-on-Positive-Loss Function

To resolve the issue coming from the imbalanced dataset during the training, focal loss [21] for dense objects was proposed. The focal loss lightens the loss-weights on dense objects. However, a cross-entropy, which weights a specific class when calculating the cross-entropy loss, was developed as a relatively balanced learning method. Here, we implemented an integrated-on-positive-loss function (IPLF), which is a more effective way to train deep-learning algorithms with imbalance datasets:(1)$$\begin{array}{cc} L(y,\hat{y}) & =y\log(1-{\left(\alpha +\frac{1}{\hat{y}}\right)}^{2}\log\left(\hat{y}\right))\\ \; & -{\left(\beta \hat{y}\right)}^{2}\left(1-y\right)\log\left(1-\beta \hat{y}\right) \end{array}$$
where $\hat{y}$ and *y* are probabilities of the prediction and the ground truth respectively, and α and β are gradient-weight constants. *y* is a linear distribution from zero to 1. $\hat{y}$ is a value one of 0 and 1. The values of α and β are set to 0.5 and 0.95, respectively. It illustrates the selection process for α and β in the discussion section. To effectively train the sparse class from imbalanced datasets, the IPLF modifies the gradients of the trainable variables by decreasing the value of the loss function for a dense class while increasing the value of a loss function for a sparse class. Therefore, for the sparse class as an RCT, the gradients are increased by the IPLF. By contrast, for a dense class as a background object, the overall gradients become lower by the IPLF.

As the primary task in this study is pixel-wise binary classification, the value of ground truth can only be a zero or one. As shown in Figure 2a,c, the value and gradient of the IPLF, which are illustrated as a black line, are lower than those of other loss functions. Thus, the IPLF slowly optimizes the deep learning model when the values of the ground truth of the easy samples are zero because smaller penalties are added to the incorrectly predicted values with smaller gradient values.

By contrast, as shown in Figure 2b,d, the value and gradient of the IPLF are higher than those of other loss functions when the values of the ground truth of the sparse samples are 1. Please note that in Figure 2d the absolute values of the gradient of the IPLF are higher than those of the others. Therefore, the IPLF optimizes the deep-learning model to shrink for the positive area of the RCT because larger penalties are added to the incorrectly predicted values with higher gradient values. As a result, a comparison graph of the values and gradients demonstrates that the IPLF provides a higher effect to optimize the positive area of the RCT and a lower effect to optimize the negative area. Thus, the IPLF can focus more on revealing the positive area of the RCT rather than the background in comparison to other loss functions.

### 2.3. Architecture of the SMART-CA

We proposed SMART-CA for a better segmentation of RCT in ultrasound images by resolving the problems caused by the low resolution and speckle noise of US images. SMART-CA adopts an encoder-decoder structure mainly used in semantic segmentation deep learning models such as U-Net and SegNet. It consists of a dual encoder architecture by adding a pre-trained encoder to an encoder part. In a previous study [23], the pre-trained architecture was utilized to effectively extract a specific feature map [24]. Therefore, SMART-CA was designed to use the pre-trained architecture to extract specific feature maps related to an RCT. The SMART-CA consists of three key components: (1) a pre-trained encoder, (2) trainable encoder, and (3) decoder. The pre-trained and trainable encoders have the same structure, including combined multiple convolution operations and activation functions. The variables in the trainable encoder are optimized to accurately predict the RCT. By contrast, the pre-trained encoder only has constant variables but is not trained during the training of SMART-CA. However, pre-trained and trainable encoders do not share any variables and have arbitrary variables. Here, the architecture of the pre-trained encoder was constructed based on VGG19, which is commonly used in a classification task. More detailed architectures of SMART-CA are illustrated in Appendix A
Table A1 and Table A2.

Furthermore, Figure 3 illustrates the performance of the segmentation of SMART-CA along with the degree of optimization of a pre-trained encoder. As shown in Figure 3, the segmentation performance was improved as the classification performance of the pre-trained encoder was improved. Therefore, the pre-trained encoder, which can generate the best performance, (Figure 3) is utilized to construct the architecture of SMART-CA.

The architecture of VGG19 was used as a baseline architecture for the pre-trained encoder because SMART-CA requires a classification task in addition to a segmentation task. Although the architecture of an encoder part of the U-Net is generally used for a segmentation task, we here utilized the architecture of VGG19 for SMART-CA since the performance of SMART-CA for the segmentation of RCT was degraded when the architecture of U-Net [25] was used as the pre-trained encoder instead of VGG19 [26], the performance of SMART-CA for the segmentation of RCT was degraded. Table 1 shows a comparison of the segmentation performance of SMART-CA with different encoder and decoder architectures. As shown in Table 1, SMART-CA offered the highest Dice coefficient of 0.771 when VGG19 and U-Net were used as baseline architectures of the encoder and decoder, respectively. VGG19 and ResNet151 [27] are common classification architectures. Therefore, when they were implemented in the decoder part of SMART-CA, the overall structure was reversed, and a maxpooling operation was replaced with a deconvolution operation.

### 2.4. Training of SMART-CA

The pipeline used for training SMART-CA is illustrated in Figure 4. Figure 4a illustrates the training pipeline of the pre-trained encoder. By solving the classification tasks that determine the existence of the RCT in US images, the variables of the pre-trained encoder are optimized to find the RCT. In other words, the pre-trained encoder is optimized with a training set for a binary classification task, which classifies the existence of the RCT in US input images (class 0, No RCT; class 1, RCT).

Figure 4b illustrates the training pipeline of SMART-CA using the same training set of ultrasound images. First, the pre-trained encoder is trained for a classification task, which is a binary inference of the presence of RCT regions in an input ultrasound image. Using the same method to optimize other conventional deep learning models of classification tasks, the pre-trained encoder is optimized by using the IPLF since the number of the dataset for the classification is also imbalanced as illustrated in the dataset’s description. While training the pre-trained encoder, the generated values become arbitrary values of between zero and 1 when the predictions are generated from a SoftMax layer. A value close to zero indicates that the probability of an RCT is extremely low, whereas a value close to 1 indicates an extremely high existence probability. By utilizing the distribution of the generated linear probability through the pre-trained encoder, the existence of an RCT is well predicted.

The trainable encoder and decoder of SMART-CA are optimized through a backpropagation method using ultrasound images of the same training set and the corresponding annotation images showing the RCT area. While training the trainable encoder and decoder, the generated values become arbitrary values of between zero and 1 pixel by pixel when the predictions generated from a decoder pass through a SoftMax layer. Thus, the generated predicted segmentation map includes the pixel-wise classification in the result of the existence of the RCT with the same size as the input ultrasound images. Similar to the classification task, a value close to zero indicates that the existence probability of RCT is extremely low, whereas a value close to 1 indicates an extremely high existence probability in a pixel-by-pixel manner. As a result, the fully optimized SMART-CA correctly predicts the regions where the RCT is present in such a pixel-by-pixel approach. Note that during this training phase, the pre-trained encoder is not optimized, and therefore the values of the parameters of the pre-trained encoder do not change.

### 2.5. Experimental Setup

In this study, the following experiments were conducted to verify the performance of the IPLF in an imbalanced dataset and SMART-CA with IPLF. (1) To verify the performance of the IPLF, the IPLF was compared with other loss functions in a semantics segmentation with an imbalanced dataset. (2) The performance of the SMART-CA with the IPLF is compared with that of conventional computer vision-based and deep learning-based state-of-the-art models.

For implementation of deep-learning architectures including SMART-CA and other state-of-the-art models, a public platform, Tensorflow [28], is used. During the training phase, the mini-batch size is 8; in addition, group normalization [29] is used because the batch size is too small to use a general normalization such as batch normalization [30]. Each model was trained for 250 epochs with 5-fold cross-validation. Each fold has 280 images, and the ratio of a training set, a validation set, and a test set was 3:1:1. The number of training images is 840. In addition, data augmentation techniques including horizontal flip, intensity variation, and cropping with resizing were applied to the datasets. Thus, a total of 5040 images were used for training each network. The training of the network was realized by using two Intel Xeon E5-2620v4 CPUs @ 2.1GHz and 4 NVIDIA TITAN Xp GPUs (12 GB) (Santa Clara, CA, USA).

SMART-CA and the other compared deep-learning architectures are implemented in the same environment based on Tensorflow. They are trained using the same hyperparameters. The deep-learning models are optimized using AdamOptimizer by using the stochastic gradient descent method [31] with a mini-batch size of 8. The first and second decays of the hyperparameters used in AdamOptimizer were set to 0.99 and 0.999, respectively. The learning rate is initially set to 0.001 but is reduced by half during every 40-epochs while training the deep-learning models [32]. Here, a dropout is not applied to the training of the models. A SeLU activation function is used as the activation function when a segmentation task is applied. During the process of optimizing a pre-trained encoder, a classification task is included. All hyperparameters are the same as in the segmentation task. A dropout of 0.75 is applied to the training of the pre-trained encoder. However, the dropout is not applied to the pre-trained decoder when the classification task is tested.

To examine how SMART-CA is robust to the noise of US images, SMART-CA performance in segmenting an RCT was compared under different speckle noise levels of the US images. Randomly generated speckle [33] noises were added to the original US images. Figure 5 shows example images with different levels of speckle noises.

### 2.6. Evaluation Metrics

Precision and recall (= sensitivity) were utilized as approximation metrics to assess the performance of algorithms for the segmentation of RCTs in ultrasound images. Because the predicted positive area of the RCT is relatively smaller than the background, the precision is more relevant to positive regions than the specificity (= true negative rate). Therefore, precision was utilized instead of specificity. The definitions of the precision, recall, and Dice coefficient are as follows: (2)$$\mathrm{Precision}=\frac{\mathrm{TP}}{\mathrm{TP}+\mathrm{FP}}$$
(3)$$\mathrm{Recall}=\frac{\mathrm{TP}}{\mathrm{TP}+\mathrm{FN}}$$
(4)$$\mathrm{Dice}\;\mathrm{Coefficient}=\frac{2\mathrm{TP}}{2\mathrm{TP}+\mathrm{FP}+\mathrm{FN}}$$
(5)$$\mathrm{Accuracy}=\frac{\mathrm{TP}+\mathrm{TN}}{\mathrm{TP}+\mathrm{FP}+\mathrm{FN}+\mathrm{TN}}$$
(6)$$\mathrm{Specificity}=\frac{\mathrm{TN}}{\mathrm{FP}+\mathrm{TN}}$$
(7)$$\mathrm{Balanced}\;\mathrm{Accuracy}\;\left(B.A.\right)=\frac{\mathrm{Recall}+\mathrm{Specificity}}{2}$$
where TP, FP, FN, TN indicate the true positive, false positive, false negative, and true negative between the annotations and predictions, respectively. Precision, Recall, Dice coefficient, Accuracy, and Balanced Accuracy (B.A.) were utilized to evaluate the performance of SMART-CA for a segmentation task. Also, accuracy was utilized to evaluate the performance of SMART-CA for the binary classification related to the training of the pre-trained encoder.

## 3. Results

### 3.1. Effects of IPLF in the Segmentation of RCT

Deep-learning architectures optimized using the IPLF are compared with those using other loss functions to evaluate the performance of IPLF for the segmentation of RCTs in US images. For the evaluation, precision, recall, and Dice coefficients of the deep-learning architectures are acquired. As the deep-learning architecture, U-Net [25], FusionNet [34], $S^{3}$ [35], and Hough-CNN [36] are used here. The architectures are optimized using the binary cross-entropy loss (BCE–loss), F1-Loss, focal loss [21], and IPLF, respectively.

Table 2 shows the segmentation performance of deep-learning architectures optimized using IPLF and other loss functions in terms of precision, recall, and Dice coefficient. The IPLF offers the highest precision, recall, and Dice coefficient values of 0.6693, 0.9302, and 0.7366, compared to other loss functions. The precision, recall, and Dice coefficient values of IPLF are improved by +44.29%, +8.09%, and +37.56% from those of other loss functions at S3. The improved recall value indicates that the positive predictions of the RCT become more accurate even in such imbalanced datasets. As a result, as shown in Table 2, the IPLF allows predicting the RCT more accurately than the other loss functions, demonstrating its capability for mitigating the potential issues caused by the imbalanced US images of RCT.

### 3.2. Evaluation of Performance of SMART-CA for the Segmentation of RCT in US Images

To evaluate the performance of our proposed SMART-CA for segmentation, SMART-CA is compared with other conventional computer-vision-based algorithms and CNN-based state-of-the-art models. Please note that SMART-CA has a pre-trained encoder that is optimized through the classification task. The pre-trained encoder is optimized in advance to achieve 80% accuracy on the test and validation sets.

Table 3 compares the performance of SMART-CA with that of the other models in terms of the Dice coefficient, in US images at different speckle noise level. The Dice coefficients of SMART-CA are higher than those of the other algorithms. In particular, at noise level 3 (PSNR = 10), other models show an extremely low performance in the segmentation of RCT, but SMART-CA is capable of acceptably segmenting the RCT with an improved Dice coefficient value of +43.3% and +37.7% higher than those of the computer-vision and deep-learning algorithms, respectively. These results demonstrate that SMART-CA predicts an RCT better than the other algorithms, even in noisy environments. Moreover, for a further evaluation of SMART-CA with the IPLF, we compared the performance of SMART-CA and other deep-learning models with the IPLF or other common loss functions at different speckle noise levels.

Figure 6 illustrated the graph of the quantitative analysis of the performance of SMART-CA, compared to those of other state-of-the-art models including U-Net, $S^{3}$, and Hough-CNN for segmentation of RCT. Figure 6, which is related to Appendix B
Table A3, illustrates that the precision, recall, and dice coefficient values of SMART-CA in the segmentation of RCT from the captured US images were 0.604% (+38.4%), 0.942% (+14.0%), and 0.736% (+38.6%), respectively. Furthermore, the precision, recall, and dice coefficient values of SMART-CA in the segmentation of RCT from US images with severe speckle were 0.337% (+22.5%), 0.860% (+15.8%), and 0.484% (+28.5%), respectively. The recall and Dice coefficient values are higher than other models with IPLF or other loss functions in the case of the speckle-noised environment. The detailed quantitative result of the Figure 6 is illustrated in Appendix B.

The segmentation results are illustrated in Figure 7. As shown in Figure 7, a comparison of the predicted images by SMART-CA and other state-of-the-art models such as U-Net, S3, and Hough-CNN, which are optimized with three loss functions using speckle-noised US images, is illustrated.

## 4. Discussion

### 4.1. Analysis of SMART-CA

In this study, we demonstrated that the proposed SMART-CA outperforms other state-of-the-art models in the segmentation of RCTs in ultrasound images. To enhance its performance, we incorporated a pre-trained encoder into the architecture of the SMART-CA. The pre-trained encoder was trained through a classification task which predicts the presence of an RCT in ultrasound images. The pre-trained encoder needed to be sufficiently optimized to achieve a good performance in the segmentation task because it utilizes the features that were extracted during the classification phase. However, the segmentation performance decreased when the accuracy of the pre-trained encoder in the classification task was higher than 80%. Therefore, we incorporated the pre-trained encoder, which offers 80% accuracy for the classification task, into the SMART-CA. The reason why the segmentation performance of SMARTCA is degraded by increasing the accuracy of the pre-encoder over 80% for classification tasks should be elucidated, which remains for further study.

### 4.2. Analysis of IPLF

To select the optimal hyperparameters (α and β) of IPLF for SMART-CA, the effect of hyperparameters on a training bias (gradients) was examined. In particular, the gradients of IPLF at different α and β values were investigated when the ground truth was 0 and 1. Figure 8 illustrates the gradients of IPLF at different α and β values. When the value of the ground truth is 0, the gradients values of IPLF are mainly determined by alpha. The lower α results in the slower optimization of SMART-CA. Here, two constraints were considered: (1) Lower gradient values of IPLF than focal loss. (2) Stable lower gradients for the dataset that would be predicted as Negative (background) optimizes the deep learning model with small enough epochs. Since IPLF becomes the same as focal loss when α is 1, α should be less than 1 to realize that the gradients of IPLF are less than those of focal loss. By doing so, the training bias on Negative can be reduced. Therefore, the value of α was selected to be 0.95 to achieve the best training performance using a grid search algorithm with the values in [0.50, 0.95] with increasing steps of 0.05.

On the other hand, when the value of the ground truth is 1, the gradients of IPLF are mainly determined by β. Here, similar to α, the constraint of large enough gradients was considered to increase the training bias of dataset to be predicted as positive. Therefore, the value of β was here selected to be 0.5 since the value of β at 0.5 results in the mean-gradient value of IPLF, resulting that a training bias rate on Positive (RCT) becomes higher in every probability. Therefore, with the optimal hyperparameters, SMART-CA can be better trained with a dataset, which exhibits the skewed RCT distribution (positive).

### 4.3. Feature Extraction by SMART-CA

The pre-trained encoder, which extracts features related to an RCT, provides local information to the RCT with a decoder when a segmentation task is conducted. Simultaneously, the features extracted from a trainable encoder are also transferred to the decoder. As a result, SMART-CA can exploit both pre-extracted features from the pre-trained encoder for the classification of RCT and the extracted features from the trainable encoder for the segmentation of the RCT. To verify that the different features are extracted from both encoders, the class activation map (CAM) [37] of CNNs are used to visualize the explanations of the CNNs by the encoders and decoder of SMART-CA. In Figure 9, the CAM demonstrates that different features are generated by each encoder. The extracted features from each encoder are used in the decoder pipeline to segment the RCT. As shown in Figure 9c, the overall area of the shoulder is focused on by the trainable encoder. By contrast, as shown in Figure 9d, the pre-trained encoder focuses more on the RCT area than the trainable encoder because the pre-trained architecture is already trained for the classification tasks to find the existence of the RCT in the US images. Therefore, the pre-trained encoder would be able to extract more informative features than the trainable encoder; therefore, SMART-CA with a pre-trained encoder would be able to segment the RCT despite the low resolution and noises of the US images.

In addition, the activation-based heatmap is analyzed to verify the impact of each layer on the US images. These heatmaps are generated by the layer of each Down block of Part I (see Appendix A
Table A1) of the encoders excluding the MaxPooling layers. As shown in Figure 10a, an RCT is present in the selected US images. The high-level features of the pre-trained encoder focus on the RCT area, as illustrated in Figure 10c. Similarly, it is recognizable that the low-level features of the pre-trained encoder would be able to extract the features of the RCT because the decoder, which uses the low-level features of the encoder, will be able to generate the RCT-related CAM, as shown in Figure 9d. By contrast, however, the features generated by the trainable encoder are disturbed by the noises of the US images instead of focusing on the RCT area in both high- and low-level features, as illustrated in Figure 10b. In addition, the activation of the CNNs is disturbed by the noises of the US; therefore, a lower activation will be achieved from the shoulder area. As a result, the activation maps illustrate that SMART-CA with a pre-trained encoder will enable more informative features related to the RCT area despite the low resolution and noisy environment of the US images.

### 4.4. Analysis of Results

The analysis of Table 1, which illustrates the recall generated by F1-Loss is higher than the recall generated by IPLF, is explained here. As illustrated in Table 1, F1-Loss outperformed other loss functions in terms of Recall in most cases. Since each number of *positive* = *TP* + *FN* is the same for all loss functions, and based on Equation (Equation 3), the true positive is especially more than other loss functions. It seemed that F1-Loss could predict the positive area with better performance than others. However, Table 1 illustrates precision [Equation (Equation 2)], which indicates the ratio of true positive from the predicted positives, i.e., the precisions by F1-Loss are remarkably less than precisions by IPLF. Here, since the number of *positive* is the same for all loss functions, followings can be calculated based on Equations (Equation 2) and (Equation 3):(8)$$\begin{array}{cc} TP & =\mathrm{Recall}\times \mathrm{Positive}\\ \mathrm{FP} & =\frac{1-Precision}{Precision}\times TP=\frac{1-Precision}{Precision}\times Recall\times Positive \end{array}$$

Here, since *positive* is the same for all loss functions, ratio of false positives of F1-Loss and IPLF can be calculated as follows:(9)$$FP_{F1}:FP_{IPLF}=\frac{1-0.36}{0.36}\times 0.8913:\frac{1-0.4763}{0.4763}\times 8849=1.63:1$$

Here, the result illustrates that the number of false positive generated by F1-Loss is 1.6 times more than the number of false positive generated by IPFL. The result demonstrates that F1-Loss predicts positives 1.6 times more than IPLF, i.e., it can be concluded that recall, the true positive rate, was higher since F1-Loss simply predicts more positives than IPLF. Furthermore, most of the state-of-the-art models, which are optimized using BCE or a focal loss, manifest coarse segmentation results with a void representation of an RCT. Conversely, the IPLF produces better-predicted RCT regions than other common loss functions, particularly in the case of implementation into SMART-CA. As the baseline architecture for SMART-CA, a combination of VGG19 and U-Net was used. However, for the further improvement of SMART-CA for diagnosis of the RCTs, other state-of-the-art models, which can offer better performance than only the combination of VGG19 and U-Net, should be tested. Furthermore, although this paper mainly focused on the noise of US images and the imbalanced problems caused by the small areas of RCTs, the other related issues including the low contrast and unclear boundaries of RCTs need to be resolved. The associated work here remains a future study.

## 5. Conclusions

In this study, we developed a novel deep-learning architecture, denoted as SMART-CA, using the IPLF and a pre-trained encoder to diagnose a location of RCT with imbalanced and noisy ultrasound images. In particular, the IPLF enhanced the segmentation performance of SMART-CA by resolving the potential issues caused by an imbalanced dataset. In addition, the pre-trained encoder allowed an accurate extraction of the feature maps related to the RCT in the noisy ultrasound images. Our experiments demonstrated that SMART-CA outperforms other state-of-the-art models for the segmentation of RCTs from ultrasound images with both normal and severe speckle noise. The precision, recall, and Dice coefficient values of SMART-CA in the segmentation of RCT from the ultrasound images were 0.604% (+38.4%), 0.942% (+14.0%), and 0.736% (+38.6%), respectively. With better performances than other state-of-the-art algorithms, the proposed SMART-CA can be applied to the diagnosis of various diseases besides RCTs. The precision, recall, and Dice coefficient values of SMART-CA in the segmentation of RCT from ultraound images with severe speckle noise were 0.337% (+22.5%), 0.860% (+15.8%), and 0.484% (+28.5%), respectively. These enhanced performances indicate that the proposed SMART-CA and IPLF can still work well despite of severely noisy environments than other algorithms. In order to improve SMART-CA’s better diagnosis of RCTs, other state-of-the-art models, which can offer better performance than the current simple combination of VGG19 and U-Net, should be tested as a baseline. The associated work here remains to be further investigated.

## Figures and Tables

**Figure 1 sensors-21-02214-f001:**
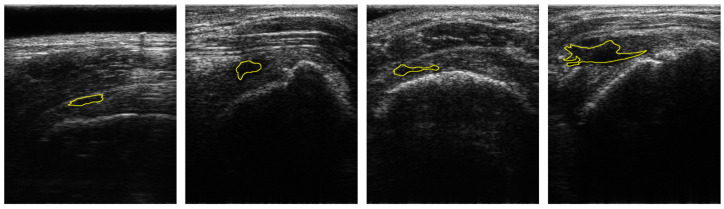
Samples of US images and the ground truths of Rotator-Cuff Tears. The ground truths are overlaid as a yellow line. Using the system of which pitch size is 0.3 mm, aperture size is 4 mm × 38 mm, capable frequency is 5 to 14 MHz, and imaging frequency for the clinical experiment is 13.3 MHz.

**Figure 2 sensors-21-02214-f002:**
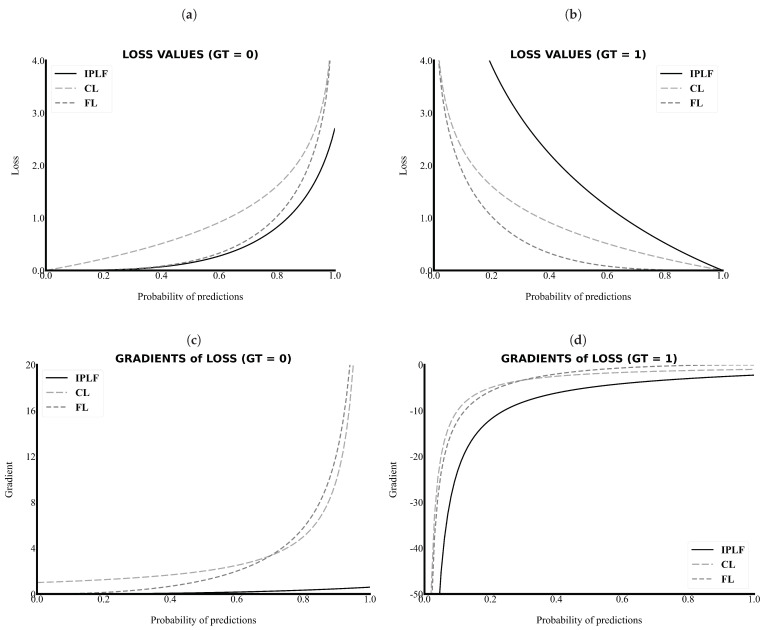
The loss values and gradients along the probability of predictions. The IPLF has a lower value and gradient in the case of the background (GT = 0) and a higher value and gradient in the case of the RCT (GT = 1). (**a**) Loss versus Probability of predictions when the ground truth is 0, (**b**) Loss versus Probability of predictions when the ground truth is 1, (**c**) Gradient versus Probability of predictions when the ground truth is 0, and (**d**) Gradient versus Probability of predictions when the ground truth is 1. IPLF, CL, FL indicate integrated-on-positive-loss function, cross-entropy Loss, and focal Loss, respectively.

**Figure 3 sensors-21-02214-f003:**
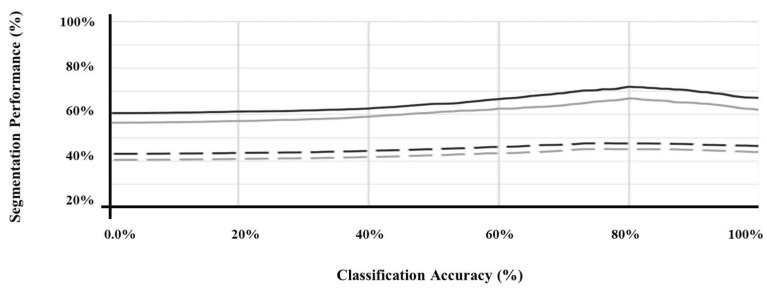
The segmentation performance of SMART-CA according to the classification accuracy of pre-trained encoder. U-Net (black) and FusionNet (gray) using the original ultrasound image (straight line) and with speckle noise of which PSNR is 15 db (dashed line).

**Figure 4 sensors-21-02214-f004:**
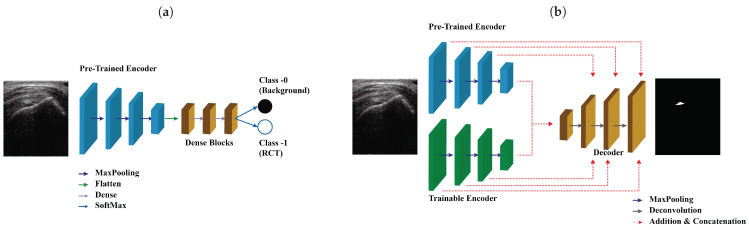
The pipeline of the training SMART-CA. SMART-CA consists of three parts; a pre-trained encoder, a trainable encoder, and a decoder. (**a**) The pre-trained encoder trained by solving the classification task that determines the existence of the RCT in US images. Then, (**b**) the trainable encoder and decoder are trained by solving the segmentation task that localizes the RCT. Please note that the pre-trained encoder is optimized in (**a**) not (**b**).

**Figure 5 sensors-21-02214-f005:**
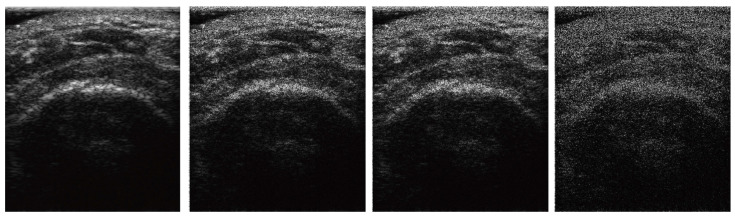
The images at different speckle noise level. Each column indicates original, 15 db, 12 db, and 10 db images.

**Figure 6 sensors-21-02214-f006:**
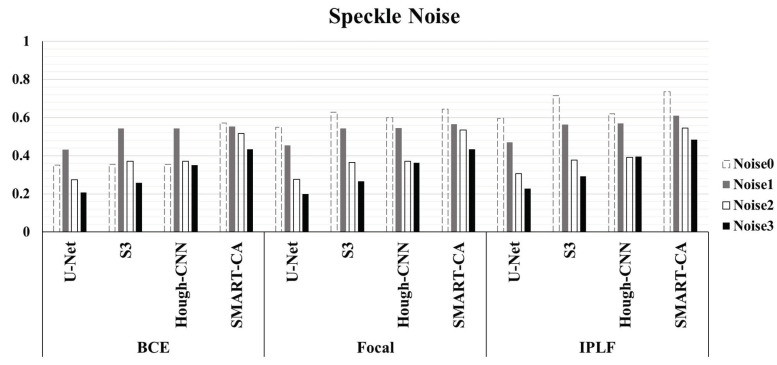
The evaluation of Dice Coefficient using different deep-learning models and different loss functions using randomly generated speckle-noised US images. In most cases, the performance of SMART-CA with IPLF shows the best performance among them despite the noisy environment of US images.

**Figure 7 sensors-21-02214-f007:**
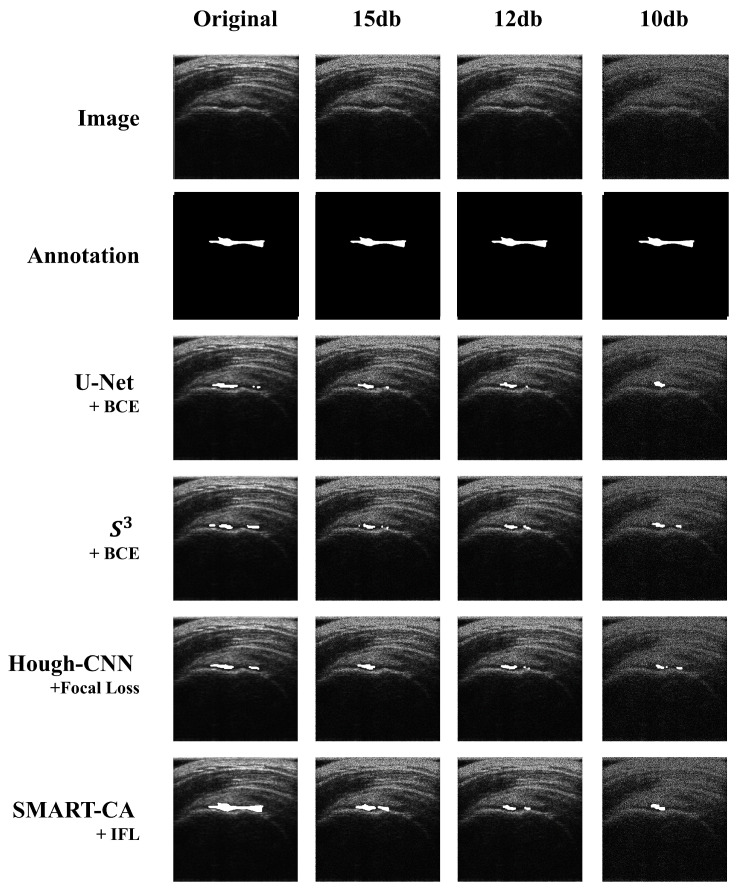
Comparison of the segmented results obtained by SMART-CA (Ours) to other state-of-the-art models including U-Net, $S^{3}$, and Hough-CNN according to the speckle-noised images.

**Figure 8 sensors-21-02214-f008:**
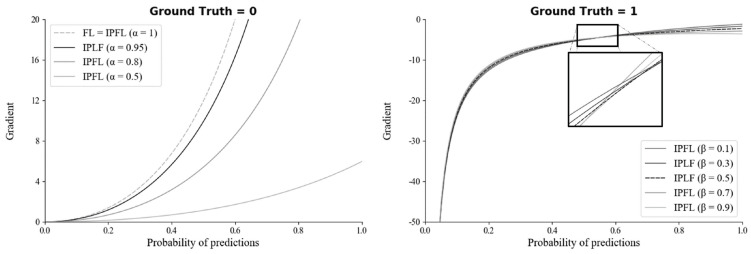
The gradients of the IPLF for each hyper-parameter of α and β when ground truth is 0 (**left**) and 1 (**right**).

**Figure 9 sensors-21-02214-f009:**
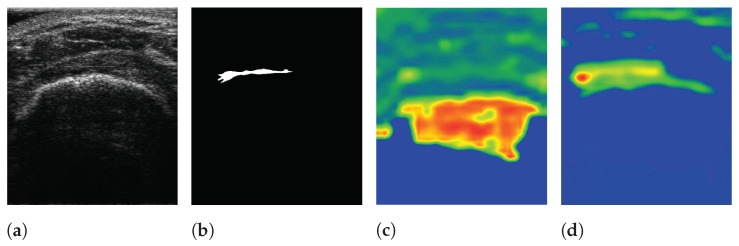
The visualization of data using SMART-CA and Class Activation Map (CAM). (**a**) raw input image, (**b**) ground truth, (**c**) CAM generated by the trainable encoder, and (**d**) CAM generated by the pre-trained encoder of SMART-CA. The red color indicates the higher attention from the CNN-based architecture and the blue color indicates the lower attention.

**Figure 10 sensors-21-02214-f010:**
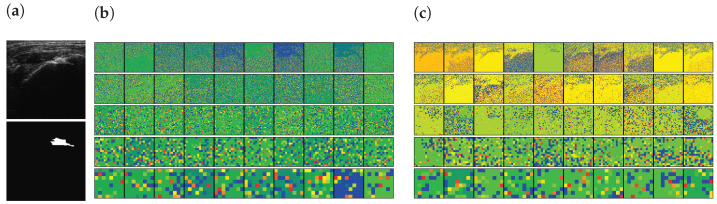
The activation-based visualization of each channel of each layer. (**a**) raw input image and ground truth of RCT, (**b**) activation-based heatmap by the trainable encoder, and (**c**) activation-based heatmap by the pre-trained encoder. The first row, second, third, fourth raw are generated by Down1, Down2, Down3, and Down4. The last row is generated by the Flattened layer. The red color indicates the higher attention from the CNN-based architecture and the blue color indicates the lower attention.

**Table 1 sensors-21-02214-t001:** Dice coefficients of SMART-CA with different baseline architecture of encoder and decoder parts in the segmentation of RCTs.

Baseline Model		Encoder						
		**VGG19**	**Google** **Net**	**ResNet** **151**	**U-Net**	**SegNet**	**Dense** **Net**	**Fusion** **Net**
Decoder	VGG19	0.525	0.537	0.504	0.513	0.536	0.525	0.527
	ResNet151	0.568	0.592	0.572	0.536	0.581	0.478	0.498
	U-Net	0.771	0.702	0.672	0.625	0.691	0.718	0.325
	SegNet	0.770	0.769	0.684	0.655	0.733	0.692	0.657
	FusionNet	0.750	0.761	0.659	0.678	0.492	0.632	0.655

**Table 2 sensors-21-02214-t002:** Comparisons of IPLF and other loss functions at different baseline architectures in terms of various evaluation metrics. The highest values are highlighted as **bold**.

		Binary Cross-Entropy Loss	F${}_{1}$-Loss	Focal Loss	Integrated on Positive Loss Function
U-Net	Precision	0.2304	0.3600	0.4075	**0.4763**
	Recall	0.8553	**0.8913**	0.8480	0.8849
	D.C.	0.3630	0.5129	0.5505	**0.6192**
	Accuracy	0.6679	0.8127	0.8468	**0.8796**
	B.A.	0.7499	0.8471	0.8473	**0.8819**
FusionNet	Precision	0.2106	0.3238	0.4009	**0.5129**
	Recall	0.9040	**0.9054**	0.8690	0.9041
	D.C.	0.3417	0.4770	0.5486	**0.6545**
	Accuracy	0.6145	0.7749	0.8418	**0.8944**
	B.A.	0.7412	0.8517	0.8537	**0.8986**
$S^{3}$	Precision	0.2264	0.3149	0.5260	**0.6693**
	Recall	0.8911	**0.8975**	0.8459	0.8190
	D.C.	0.3610	0.4663	0.6487	**0.7366**
	Accuracy	0.6511	0.7726	0.8986	**0.9352**
	B.A.	0.7562	0.8273	0.8755	**0.8843**
Hough-CNN	Precision	0.2326	0.3138	0.4767	**0.5021**
	Recall	**0.8761**	0.8592	0.8493	0.9302
	D.C.	0.3676	0.4597	0.6106	**0.6521**
	Accuracy	0.6665	0.7765	0.8802	**0.8902**
	B.A.	0.7583	0.8127	0.8667	**0.9077**

**Table 3 sensors-21-02214-t003:** The performance of SMART-CA with IPLF and other models at different speckle noise levels. The highest Dice Coefficient values are highlighted as **bold**.

Speckle Noise	Watershed	Active Contour	U-Net	$S^{3}$	Hough-CNN	SMART-CA
Original	0.562	0.636	0.595	0.715	0.621	**0.736**
PSNR = 15	0.406	0.486	0.470	0.563	0.569	**0.609**
PSNR = 12	0.168	0.224	0.307	0.378	0.392	**0.546**
PSNR = 10	0.051	0.107	0.227	0.292	0.395	**0.484**

## Data Availability

Not applicable.

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
