# Peer review of "Imbalanced Loss-Integrated Deep-Learning-Based Ultrasound Image Analysis for Diagnosis of Rotator-Cuff Tear"

_sensors, 2021, doi:10.3390/s21062214_

Round 1

Reviewer 1 Report

Summary: The novelty of this study is to propose the SMART-CA network and the positive loss function. The authors demonstrated the performance of the loss function as well as the new network, in comparison to other loss functions and conventional approaches. The results show the improved results, compared to the references, but the level of performance is less than 50% and not enough for clinical application, which justifies the need of further improvement in the future work. Regardless of the novelty of this study, there is still room to improve the structure of this manuscript to help prospective readers understand this study clearly.

Major comments:

  • Based on the abstract, it seems that the novelty of this study includes (1) adopting a pre-trained classification architecture (SMART-CA) and (2) developing the positive loss function (IPLF). However, from the last paragraph of the introduction, the authors said, “we propose a convolutional neural network (CNN) based deep-learning architecture, ...” Thus, the novelty of this study should be clarified and consistent across the manuscript.
  • The meaning of “imbalanced” should be clarified.
  • Throughout the manuscript, methods/results/discussion were mixed up and hard to be separated. Therefore, I strongly recommend the authors to reorganize each subsection, as suggested below, separating the methods/results/discussion clearly.

Specific comments:

Abstract

the positive loss function (IPLF) should be used, instead of a novel imbalanced loss function (IPLF).

Introduction

P2, lines 50-52: “Furthermore, deep-learning-based RCT diagnostic analysis has been developed with an improved performance achieved through the development of image processing techniques”. Does it mean that DL-based approaches outperform conventional image processing techniques?

P2, lines 53-64: Good motivation for this study. However, not sure about this rationale: “RCT regions in an ultrasound image are much smaller (typically less than 10%) than non-diseased areas.” The performance of DL-based segmentation (or detection) is limited by the percentage of a target region? 10% may be not too small to be detected by using DL.

What is the meaning of an “imbalanced” dataset? Which means that US images are not consistent across subjects or regions due to operator-specific variations? Please clarify the meaning of “imbalanced” in terms of US images.

Methods

2.1. Cross validation (eg, 5-folded) may be a good option for future work, because the dataset is small.

Figure 1. Ground truths are shown here, but it would be better to indicate overlaid areas in the US images also, for example, by using dotted lines or etc, because it’s hard to recognize the RCTs in the US images.

“However, the datasets for RCT are imbalanced because the regions of the RCTs were small within the field of view (FoV) of the ultrasound images.” I can’t understand why small RCTs cause the “imbalance” which has not been clearly explained so far.  

“This demonstrates that the dataset is imbalanced with the small area of the RCT.” Same as above. Also, instead of “demonstrates”, it seems more appropriate to use “illustrates” here.

What is the meaning of “biased training”? High “bias” means that prediction is far from ground truth, but the dataset itself was annotated by two expert physicians, although there may be operator-dependent variation, and the “manual” segmentation was considered accurate.

2.2. I am still confused about “imbalanced” vs. “balanced”. Why is a cross-entropy called “a balanced cross-entropy” here?

“because the previously proposed loss functions were developed for detection and classification tasks, they are not suited for semantic segmentation tasks.”

I can’t agree with this statement, if the authors focus on the “segmentation” of RCTs by using a CNN. The most-widely used U-Net was proposed for segmentation that is a binary pixelwise “classification”, isn’t it? L2 loss is the most widely used loss function. Please refer the following. https://arxiv.org/abs/1505.04597 I am confused at this point. This study is focused on “classification” or “segmentation” or both? Figure 1 gives the impression that this study is focused on “segmentation” but the title (diagnosis) and this section give another impression of “classification (with or without a RCT, simple binary question)”.

In general, y_hat is considered “prediction”.  

“the segmentation task in this paper is the binary classification”. This is a very confusing statement. Does it mean that the authors solved the “segmentation” problem by this pixelwise “binary classification” approach? Please clarify.

“the IPLF slowly optimizes the deep learning model when the values of the ground truth of the easy samples are zero because smaller penalties are added to the incorrectly predicted values with smaller gradient values.” This is not clear to me... what are “easy samples”? background? Also, “sparse samples” seem to indicate “RCTs”. But I am still confused about the usage of these terms. Please use the terms consistently throughout the manuscript for clarity.

Figure 2. Please use a normal straight line for IPLF to distinguish it from the others.

2.3. What does “SMART-CA” stand for? Did the authors propose this architecture for the first time? It seems that the proposed network (Figure 3b) is a U-Net like architecture, as mentioned in the manuscript.

Please emphasize the novelty of the proposed network in the abstract and the introduction: “dual encoder architecture by adding a pre-trained encoder to an encoder part”. Until reaching at this point, no clear statement was found about the novelty of this architecture.

Figure 3 clarified all the confusing statements above. Please clarify the authors implemented two networks in the abstract and introduction, briefly: one for classification to prepare a pre-trained encoder and the other for segmenting RCTs. It would be better to show 2.3 before 2.2, to avoid any confusion.

2.4. Why is the IPLF not used for the pre-training? Please explain.

2.5. Lines 195-199. “The pre-trained encoder, ...  trainable encoder for the segmentation of the RCT.” Is the novelty of this study, which should be emphasized in the Abstract and the Introduction. Figure 4 is Great. The correlation between Figure 4 and Figure 5 is reasonably interpreted.

Line 224: “demonstrate” à “illustrate”

Results

3.1 and 3.2 should be moved to the Method. Also, 2.5 is like to discuss why this network work, which would be good to be move to the Discussion.

3.1. Considering the data small size (only 1,400 images), it seems that the training period of 4 days is too long... is this because of too much data augmentation? Did it take 250 epochs to achieve the convergence of the loss curve?

3.2. Please clarify that the evaluation metrics are for the binary classification, not for segmentation.

3.3. In Table 1, I agree that IPLF outperforms for the precision and DC, not for “recall” that F1-loss works better. Please discuss this in the Discussion. Also, what is the accuracy (% of correction prediction) of each method?

3.4. “Moreover, to examine how robust SMART-CA is to the noise of US images, the performance of SMART-CA in segmenting an RCT is compared under different speckle noise levels of the US images. Randomly generated speckle [35] noises are added to the original US images. Fig 6 shows example images with different levels of speckle noises.” Good approach, but this should be a part of the Method.

“noise level 3” means PSNR = 10? Please clarify and be consistent.

Table 2 and Table 3 can be combined if Table 3 includes the contents of Table 2. No overlap is allowed. Also, I want to recommend to show Figure B1 in the manuscript and move Table 3 to the Supplementary document.

“The recall and Dice coefficient are much higher than those of the other models with the IPLF or other loss functions. The recall and Dice coefficient values are “much higher” than those of the other models with the IPLF or other loss functions in the speckle-noised environments.”

Please do not use “much higher” and quantify the relative or absolute improvement. When comparing values, the amount of improvement is very relative...

“Most of the state-of-the-art models, which are optimized using BCE or a focal loss, manifest coarse segmentation results with a void representation of an RCT. Conversely, the IPLF produces better predicted RCT regions than other common loss functions, particularly in the case of implementation into SMART-CA. Furthermore, SMART-CA exhibits noise-rebellious properties while predicting RCT regions. The experimental results demonstrate that IPLF can compensate the attention to the positive RCT area, and SMART-CA exhibits a precise prediction despite the noisy environment of US images.” This should be moved to the Discussion.

In Figure 3, all the methods show similar results on the left; while, on the right, the proposed SMART-CA with IPLF outperforms the other methods only for the original and 15db cases. Please discuss this point specifically. Also, please explain why the authors selected theses two cases.

Discussion

4.1. Figure 8 should be explained in the Method first and then discussed here, though the understanding of this result is planned as a future work.

4.2. Same as above. Table 4 justifies why the authors had chosen the proposed architecture, thus this should be clearly stated in the Method first, and then, if necessary, discussed here again.

4.3. Same as above.

The last paragraph should include the “limitations” and potential future work of this study.

Conclusion

“The precision and Dice coefficient values of SMART-CA in the diagnosis of RCT were 0.337% and 0.484% ... the proposed SMART-CA can be applied to the diagnosis of various diseases using US images.” I can’t agree with this point because the precision less than 50% can be applied to the diagnosis task in clinical practice. This study just shows the potential and a little improved performance, compared to the performances of conventional and other DL methods.

Author Response

We thank you for the reviewers' comments and modify the manuscript based on the comments. The comments would have been improved the quality of the manuscript. 

Reviewer 2 Report

The current paper aims to perform segmentation within ultrasound images, in order to detect the Rotator Cuff Tear (RCT) affection. An original CNN architecture, called  segmentation model adapting pre-trained classification architecture (SMART-CA), was developed for this purpose. This architecture involved an encoder part, formed by a pre-trained encoder, besides a trainable encoder, as well as an integrated on the positive loss function (IPLF), designed for imbalanced datasets, in order to focus on the RCT regions. Concerning the encoder, an architecture  based on VGG19 provided the best results. The proposed method was appropriately experimented, for different levels of specific noise. The article is well written, of a good technical and scientific quality, however, the following remarks should be taken into account:

  • the state of the art should be presented in a more extended manner, so more scientific approaches assuming the automatic diagnosis within ultrasound images should be described
  • the sensitivity of the method to changes in illumination and resolution should be discussed
  • the resulted performance can be improved, so more CNN architecture, such as GoogleNet and DenseNet, should be experimented for both the encoder and decoder parts
  • the Intersection over Union (IoU) should be also taken into account as a performance evaluation metric, for a better assessment of the proposed method
  • there are some English language issues within the text that should be taken into account:  for example, the phrase "Table. 2 compares the performance of SMART-CA as compared with that of the other models" should be replaced by  "Table. 2 compares the performance of SMART-CA with that of the other models".

Author Response

(The authors gave the same response as above.)

Round 2

Reviewer 1 Report

The authors improved the manuscript and a few minor edits are recommended before publication.

  1. " the background area occupies 9 times than the RCT area". I agree that the ratio of the areas is important. However, the contrast of the areas (or boundaries between RCTs and background) is also important but not mentioned or discussed yet. Based on Figure 1.
  2. Formula (8)-(10) are redundant. They are shown in the Method.
  3. For the "pre-trained network", please refer this paper: https://arxiv.org/abs/1603.08155.
  4. Conclusion: "The proposed SMART-CA can be applied to the diagnosis of various diseases 423 besides RCTs." As pointed out in the previous review, this statement is not appropriate based on the low accuracy of this technique. For clinical application, its accuracy should be comparable to the accuracy provided by operators. The conclusion is too long. Please simplify it.

Author Response

Thank you for the reviewers' further comments. 

we modified the manuscript based on the comments, and it would be helpful to improve the manuscript.

Thank you again.
